# Gen-MURE: Generalized Multiplicative Unbiased Risk Estimate

## Abstract

Coherent imaging modalities such as ultrasound and synthetic aperture radar (SAR) images are degraded by signal-dependent multiplicative noise, where the noise distributions vary widely across acquisition scenarios. Existing self-supervised image denoising methods either assume zero-mean additive noise, independence across pixels or require the noise distribution to be known, which often limit their applicability in real-world image denoising systems. We propose a Generalized Multiplicative Unbiased Risk Estimate (Gen-MURE), a model-agnostic self-supervised image denoising framework for enhancing images corrupted by signal-dependent, multiplicative speckle. Gen-MURE does not rely on explicit assumptions of the exact noise distribution and formulates a principled framework capable of denoising speckle ranging from different distributions. Gen-MURE does not require access to clean ground truth images or parameters of the noise model, and denoise images in one single step without any iterative refinement. Extensive experiments on ultrasound images along with unseen simulated and real SAR images demonstrate the efficiency and robustness of Gen-MURE. The code will be published upon acceptance.

## 1 Introduction

Image denoising has been studied extensively for the past several decades, with applications in biomedical engineering, remote sensing, and astronomical images, among other scientific domains. In these applications, the objective is to recover the high fidelity image from measurements, corrupted by environmental interference, sensor noise, and acquisition artifacts. The existing enhancement algorithms improve image quality by reducing noise while preserving edges and other structural details. Deep learning algorithms have become extremely successful in scientific fields involving image analysis, image enhancement, and denoising applications, and have become state-of-the-art methods. However, most of these algorithms require ground truth reference images during training, which becomes a bottleneck in several scientific applications where a noiseless reference image is impossible to obtain.

In recent years, self-supervised learning methods have emerged as state-of-the-art methods for image restoration problems where clean, ground-truth images are unavailable. Learning-based self-supervised denoising methods need the noise distribution to be known, such as SURE (Stein, 1981), Noisier2Noise (Moran et al., 2020) or assume that the noise is independent across pixels, such as Noise2Self (Batson & Royer, 2019). The performance of SURE-based algorithms is severely affected when the noise distribution is misspecified. However, the independence across noise pixels might not hold specifically in real-world applications where noise can be highly complicated.

In coherent imaging modalities, noise is signal-dependent and multiplicative in nature. Additionally, depending on the specific application, the noise can be modeled as different distributions. For example, single look Synthetic Aperture Radar (SAR) images, the noise can be modeled as the exponential distribution where the signal-to-noise ratio (SNR) is very low. On the other hand, as the number of looks increases for SAR, the noise can be modeled as the gamma distribution. For other coherent modalities, the signal-dependent noise can also be modeled as samples from the Rayleigh or Rician distributions. In practical image denoising scenarios, the exact noise distribution might not be accurately known and the zero-mean corruption assumption

made in several denoising algorithms does not hold. While most of the self-supervised frameworks have been proposed for zero-mean Gaussian noise, several advances have been made for multiplicative noise, Poisson noise and other noise distributions. However, existing algorithms do not provide a generalized approach for scenarios where the exact noise distribution is unknown. The proposed denoising method addresses this gap by introducing a robust and principled self-supervised single-image denoising framework.

The contributions of the work are summarized as follows: **(i)** We propose Generalized Multiplicative Unbiased Risk Estimate, **Gen-MURE**, a model-agnostic image denoising framework for images corrupted with multiplicative noise, without any explicit parametric assumptions about the underlying noise model. **(ii)** We develop a self-supervised, single-step image denoising formulation by leveraging local statistics without requiring access to noise parameters or clean, ground truth images. **(iii)** We extensively evaluate the performance of Gen-MURE for speckle following gamma and Rayleigh distributions and further evaluate its generalization capabilities under distribution and modality shifts. **(iv)** Extensive experiments on ultrasound images and previously unseen real and simulated data validate the denoising capabilities and robustness of Gen-MURE compared to state-of-the-art image denoising baselines.

## 2 Related Work

### 2.1 General Noise Models

**SURE:** Stein's Unbiased Risk Estimate (SURE) (Stein, 1981) was proposed to enhance noisy images, assuming that the noise distribution is known (Efron, 2004). The formulation has been adopted for several other noise distributions, such as exponential noise (Hudson, 1978), Poisson noise (Luisier et al., 2010), and Poisson-Gaussian mixed noise (Le Montagner et al., 2014). An unsupervised, nonparametric empirical Bayes least-squares framework is proposed in (Raphan & Simoncelli, 2011) as a generalized framework for SURE. Based on the SURE formulation, unsupervised learning frameworks have been proposed in (Metzler et al., 2018; Chen et al., 2022) which assume the noise distributions to be known. Ensemble-SURE (ENSURE) (Aggarwal et al., 2022) was proposed for training deep image reconstruction algorithms with noisy, undersampled images, where the measurement operator is randomly chosen from a given set. In (Zhussip et al., 2019b), the authors present a deep learning framework based on a variant of SURE for denoising undersampled images with ground truths or image priors. Another method, eSURE (Zhussip et al., 2019a) for training denoising frameworks with correlated pairs of noisy images, which can also be applied to pairs of uncorrelated images corrupted with Gaussian noise. All of these methods at least assume that the noise distribution or the noise parameters are known at the time of denoising, which is often a bottleneck in real-world applications.

**Noise2x:** Noise2Noise (Lehtinen et al., 2018) presents a self-supervised image denoising framework when two independent, zero-mean, additive noisy realizations of the same image are available. However, acquiring two independent noisy images corresponding to the same original image is often a bottleneck in real-world scenarios. To address this, Noisier2Noise (Moran et al., 2020), Recorrupted2Recorrupted (R2R) (Pang et al., 2021) and the coupled bootstrap method (Oliveira et al., 2024) have been proposed, where independent noisy realizations are obtained for any image by adding synthetic noise. Noise2Score is a self-supervised image denoising framework proposed in (Kim & Ye, 2021), which learns the score function directly from noisy images and has different post-processing steps for different noise distributions. In (Kim et al., 2022), the authors improve the Noise2Score formulation and introduce an algorithm to estimate the noise distribution and the model parameters. Noise2Void (Krull et al., 2019) denoises single noisy images employing blind-spot networks, which first mask the target pixels and then predict them from the surrounding neighborhoods. In Noise2Self (Batson & Royer, 2019), the authors propose a more general denoising framework that assumes the statistical independence of noise in different measurement directions. Probabilistic Noise2Void (Krull et al., 2020), dilated blind spot networks (Wu et al., 2020) and Laine19 (Laine et al., 2019) further improve self-supervised single image denoising algorithms.

Algorithms formulated on SURE frameworks require complete knowledge about the noise distribution, whereas denoising frameworks based on the Noise2x make minimal assumptions about the noise distribution. Although the noise distribution is never completely known in practical denoising applications, frameworks like Noise2Noise often fail to match the performance of supervised algorithms. UNSURE (Tachella et al.,

2024) proposes a theoretical framework similar to SURE, but overcomes the requirement of knowing the noise parameters during denoising. Most of these algorithms are proposed for additive noise models. Extending these frameworks to signal-dependent, non-additive noise models is challenging, specifically in applications where the noise characteristics need to be inferred directly from the images.

## 2.2 Multiplicative Noise Models

Recent works have addressed the challenges of reducing multiplicative noise without requiring clean ground truth images. Multiplicative noise, speckle, is observed in all coherent imaging modalities and is signal-dependent. Multiplicative Unbiased Risk Estimator (MURE) (Seelamantula & Blu, 2015) was formulated similar to SURE, specifically considering a gamma noise distribution. The lognormal unbiased risk estimator (LURE) (Bakshi et al., 2025) shows that applying a logarithmic transformation to multiplicative noise and subsequently using additive Gaussian denoising algorithms leads to suboptimal performance compared to methods developed directly for multiplicative noise. However, both MURE and LURE are Monte Carlo-based methods, and the inference step can be computationally expensive, thus limiting their deployment for large-scale real-time denoising. In (Gupta et al., 2025b), the authors improve the MURE formulation and present a Monte Carlo-based step to estimate the modified MURE objective function with deep networks. Several self-supervised formulations similar to blind-spot networks have been proposed specifically to reduce speckle in recent years. Speckle2Void (Molini et al., 2022) adapts the Noise2Void strategy to reduce speckle in SAR images. Speckle2Speckle (Göbl et al., 2022) is proposed to improve the quality of ultrasound images by using multiple noisy realizations of the image tissue structure. Collectively, these methods demonstrate the effectiveness of principled denoising frameworks involving supervised and unsupervised learning strategies towards reducing multiplicative noise.

MURE and LURE are computationally expensive, whereas Speckle2x frameworks have minimal assumptions that may limit their performance. In this work, we propose a generalized self-supervised denoising framework for reducing multiplicative noise corresponding to different noise distributions.

## 2.3 Formulations of SURE and MURE

First, we will briefly review Stein's Unbiased Risk Estimate (SURE) (Stein, 1981) for additive noise and then for Multiplicative Unbiased Risk Estimate (MURE) (Seelamantula & Blu, 2015) for multiplicative noise.

### 2.3.1 Additive Noise: SURE

Given an observed image $y$ degraded by additive Gaussian noise $\epsilon \sim \mathcal{N}(0, \sigma^2)$ corresponding to the true underlying image $x$, SURE provides an unbiased estimate of $\arg\min_{\hat{x}} \mathbb{E}\|\hat{x}(y) - x\|^2$ without requiring access to the unknown true signal $x$. This makes SURE an essential tool for unsupervised and self-supervised denoising and serves as a basis for several extensions for different noise distributions.

The observed images $y$ and the unobserved clean images $x$ are related as:

$$y = x + \epsilon \tag{1}$$

Since $\epsilon \sim \mathcal{N}(0, \sigma^2)$, the noisy observation $y$ is also Gaussian , i.e, $y \sim \mathcal{N}(x, \sigma^2)$ and $\sigma^2$ is the variance of Gaussian noise. Real-world applications where acquiring clean, uncorrupted signals is expensive or impossible, SURE utilizes the noisy images $y \in \mathbb{R}^N$ to minimize the mean squared error between $x \in \mathbb{R}^N$ and its estimate $\hat{x}(y) \in \mathbb{R}^N$, which cannot be evaluated directly due to the unknown $x$.

$$\mathbb{E}\|\hat{x}(y) - x\|^2 = \mathbb{E}[\|\hat{x}(y) - y\|^2 + 2\sigma^2 \operatorname{div} \hat{x}(y) - N\sigma^2] \tag{2}$$

The term $\operatorname{div} \hat{x}(y)$ represents the divergence function $\operatorname{div} \hat{x}(y) = \sum_{i=1}^{N} \frac{df_i}{dy_i}(y)$. Thus, the SURE objective can be used as a surrogate objective function, thus enabling training without access to clean images.

### 2.3.2 Multiplicative Noise: MURE

The MURE objective was proposed specifically for images corrupted with signal-dependent multiplicative noise. In coherent imaging modalities, the original unobserved image $x \in \mathbb{R}^{+\mathbb{N}}$ is degraded with multiplicative noise $\epsilon$ to create the noisy observed image $y \in \mathbb{R}^{+\mathbb{N}}$.

$$y = x \cdot \epsilon \tag{3}$$

where $\epsilon$ represents multiplicative noise and $\cdot$ represents element-wise multiplication. Depending on the acquisition modality, the multiplicative noise $\epsilon$ can follow gamma, exponential, or Rayleigh distributions. Since multiplicative noise is signal-dependent, the SURE objective given by Eq. 2 must be reformulated for multiplicative noise. MURE proposed an unbiased estimate of the mean squared error $\mathbb{E}\|\hat{x}(y) - x\|^2$ as follows (Seelamantula & Blu, 2015) for the $k$-look acquisition scenario, where $\hat{x} : \mathbb{R}^{+\mathbb{N}} \to \mathbb{R}^{+\mathbb{N}}$ is the denoising function.

$$\mathbb{E}\|\hat{x}(y) - x\|^2 = \frac{1}{N}\Big(\frac{k}{k+1}\|y\|^2 + \|\hat{x}(y)\|^2 - 2y^T \mathcal{Q}\,\hat{x}(y)\Big) \tag{4}$$

where $\mathcal{Q}$ is an operator for 1-D estimator $\hat{x}$.

$$\mathcal{Q}\,\hat{x}(y) = k \int_0^1 \hat{x}(sy)s^{k-1}ds \tag{5}$$

In practice, the MURE objective given by Eq. 4 can be used analogously to SURE, thus providing a principled extension for multiplicative noise models. A computationally efficient implementation of MURE has further been proposed in (Gupta et al., 2025a).

## 3 Method

### 3.1 Multiplicative noise formulations

We assume that images corrupted with signal-dependent multiplicative noise are observed and the true, underlying images are unknown. Images corrupted with multiplicative noise can be expressed as Eq. 6 where $x$ represents the true image and $y$ represents the image corrupted by multiplicative noise $\epsilon$ and $\cdot$ represents element-wise multiplication.

$$y = x \cdot \epsilon \tag{6}$$

Noise $\epsilon$ can be sampled from the gamma, Exponential, or Rayleigh distributions, depending on the application. In supervised learning where the paired data $(x, y)$ is known, the true image $x$ can be recovered by minimizing the supervised loss

$$\arg\min_{\hat{x}} \mathbb{E}\|\hat{x}(y) - x\|^2 \tag{7}$$

where $\hat{x}$ is the denoising function. In this work, we assume that $x$ is unobserved and we derive an unbiased estimate of Eq. 7 to recover $x$.

The supervised mean-squared error objective $\mathbb{E}\|\hat{x}(y) - x\|^2$ can be expanded as

$$\mathbb{E}\bigg[\|x\|^2 + \|\hat{x}(y)\|^2 - 2\,x^T\hat{x}(y)\bigg] \tag{8}$$

We need to replace $\|x\|^2$ and $x^T\hat{x}(y)$ with functions of the observed image $y$.

For all components of $x$,

$$\|x\|^2 = \sum_{i=1}^{N} \|x_i\|^2, \quad \text{and} \quad y_i = x_i\epsilon_i \tag{9}$$

Thus, we have

$$\mathbb{E}\left[y_i^2\right] = x_i^2 \, \mathbb{E}\left[\epsilon_i^2\right] \tag{10}$$

Assume that the multiplicative noise $\epsilon_{i=1}^N$ is independent and distributed identically according to a known distribution, $\mathbb{E}\left[\epsilon_i^2\right] = \mathbb{E}\left[\epsilon^2\right]$ for all $i$ (Seelamantula & Blu, 2015). Thus, we have

$$\|x\|^2 = \frac{\|y\|^2}{\mathbb{E}[\epsilon^2]} \tag{11}$$

Now, $y = x \cdot \epsilon$. Since $x$ is deterministic, $\mathbb{E}[y] = x\,\mathbb{E}[\epsilon]$.

Thus, $\mathbb{E}\|\hat{x}(y) - x\|^2$ can be written as

$$\mathbb{E}\left[\frac{\|y\|^2}{\mathbb{E}[\epsilon^2]} + \|\hat{x}(y)\|^2 - 2\left(\frac{\mathbb{E}[y]}{\mathbb{E}[\epsilon]}\right)^T \hat{x}(y)\right] \tag{12}$$

The self-supervised training objective is given as

$$\mathbb{E}\|\hat{x}(y) - y\|^2 = \mathbb{E}[\|\hat{x}(y)\|^2 + \|y\|^2 - 2y^T\,\hat{x}(y)] \tag{13}$$

By substituting $\mathbb{E}[\|\hat{x}(y)\|^2]$ with $\mathbb{E}\|\hat{x}(y) - y\|^2 - \mathbb{E}\|\hat{x}(y)\|^2 + 2y^T\,\hat{x}(y)$ in Eq. 12, the following expression is obtained as:

$$\mathbb{E}\left[\|\hat{x}(y) - y\|^2 + \|y\|^2\left(\frac{1}{\mathbb{E}[\epsilon^2]} - 1\right) + 2y^T\,\hat{x}(y) - 2\left(\frac{\mathbb{E}[y]}{\mathbb{E}[\epsilon]}\right)^T \hat{x}(y)\right] \tag{14}$$

**Proposition.** *An approximation of $\mathbb{E}[\epsilon^2]$ is given by $1 + \dfrac{\sigma^2}{\mu^2}$ where $\mu$ and $\sigma^2$ represent the mean and variance of the observed image $y$.*

**Proof :** Given $y = x \cdot \epsilon$,

$$\mathbb{E}[y] = x\,\mathbb{E}[\epsilon], \quad \mathbb{E}[y^2] = x^2\,\mathbb{E}[\epsilon^2] \tag{15}$$

and,

$$\mathrm{Var}(y) = x^2\,\mathbb{E}[\epsilon^2] - x^2\,\mathbb{E}[\epsilon]^2 \tag{16}$$

Therefore, $\dfrac{\mathrm{Var}(y)}{\mathbb{E}[y]^2} = \dfrac{\mathbb{E}[\epsilon^2]}{\mathbb{E}[\epsilon]^2} - 1$. In coherent imaging, speckle is commonly as multiplicative, with unit mean, i.e, $\mathbb{E}[\epsilon] = 1$. Hence,

$$\mathbb{E}[\epsilon^2] = 1 + \frac{\sigma^2}{\mu^2} \tag{17}$$

where $\mu = \mathbb{E}[y]$ and $\sigma^2 = \mathrm{Var}(y)$. $\square$

Thus, the final generalized objective for reducing multiplicative noise without clean ground-truth images can be simplified as

$$\mathbb{E}\left[\|\hat{x}(y) - y\|^2 - \|y\|^2\frac{\sigma^2}{\sigma^2 + \mu^2} + 2(y - \mu)^T\hat{x}(y)\right] \tag{18}$$

## 3.2 Noise Distributions

We consider speckle samples from two noise distributions: (i) gamma and (ii) Rayleigh. These distributions are widely used to model multiplicative speckle in coherent imaging systems. The gamma distribution is used to model multi-look speckle that is partially developed. On the other hand, Rayleigh distribution is used to model fully developed speckle.

The probability density function (pdf) of the gamma distribution with shape parameter $\alpha > 0$ and scale parameter $\theta$ is given by Eq. 19, where $\epsilon$ is the multiplicative speckle, $\alpha$ represents the number of looks, and $\gamma(\cdot)$ is the gamma function.

$$p_{\text{gamma}}(\epsilon;\, \alpha,\, \theta) = \frac{1}{\gamma(\alpha)\,\theta^\alpha}\, \epsilon^{\alpha-1} e^{\dfrac{-\epsilon}{\theta}}, \quad \epsilon \geq 0 \tag{19}$$

The pdf of the Rayleigh distribution is given by Eq. 20, where $\epsilon$ is the multiplicative speckle and $\sigma > 0$ represents the scale parameter.

$$p_{\text{Rayleigh}}(\epsilon;\, \beta) = \frac{\epsilon}{\sigma^2}\, e^{\dfrac{-\epsilon^2}{2\sigma^2}}, \quad \epsilon \geq 0 \tag{20}$$

### 3.3 Training Details

### 3.3.1 Training Objectives

Given a noisy observation $y$, the denoising network $\hat{x}_\theta$ is trained in an unsupervised manner by minimizing the training objective in Eq. 18. This recovers the clean underlying image $x = \hat{x}_\theta(y)$. To effectively guide the denoiser $\hat{x}_\theta$ to faster convergence, we incorporated an additional loss function based on the structural similarity index measure (SSIM).

Computing the SSIM for evaluating denoised images requires knowledge of the clean ground-truth image. In the absence of reference images, a modified SSIM loss is proposed to guide the denoiser to preserve structural information, without explicitly relying on clean ground truth images.

$$\mathcal{L}_{SSIM} = \text{SSIM}(\hat{x}(y), c(y)) \tag{21}$$

The modified SSIM loss is given by Eq. 21 where $y$ is the noisy observation and $c(\cdot)$ is a local smoothing operator. When applied to $y$, $c(y)$ represents a locally averaged estimate of the true underlying image. This suppresses high-frequency noise and preserves coarse structural information, making $\mathcal{L}_{SSIM}$ a structural regularizer during training.

In summary, the overall training objective is given by Eq. 22, where $\lambda$ is a hyperparameter.

$$\mathcal{L} = \mathbb{E}\left[ \|\hat{x}(y) - y\|^2 - \|y\|^2 \frac{\sigma^2}{\sigma^2 + \mu^2} + 2(y - \mu)^T \hat{x}(y) \right] + \lambda\,\text{SSIM}(\hat{x}(y), c(y)) \tag{22}$$

### 3.3.2 Training Methodology

The denoiser network $\hat{x}_\theta$ is implemented with DnCNN (Zhang et al., 2017), comprising eight feedforward blocks with batch normalization and residual learning. To improve training stability, we employ the warm-start strategy and the randomly initialized DnCNN is first trained with Noisier2Noise (Moran et al., 2020). Warm-start strategies are used in several image analysis algorithms to improve the stability and accelerate convergence during early stages of training. In the context of self-supervised training, this becomes important since clean, noiseless images are not accessible during the training paradigm. Since Noisier2Noise does not require clean ground truths or paired noisy observations, this framework is well-suited for initializing the proposed unsupervised denoising algorithm.

The training algorithms are given in Algorithms 1 and 2. After warm-starting with Noisier2Noise, the network parameters $\theta$ are trained following Eq. 18. The terms $\mu$ and $\sigma^2$ are calculated over $k \times k$ local neighborhoods instead of point estimates to obtain a robust representation of local intensity variations. Additionally, the averaging operator in Eq. 21 is implemented with a $m \times m$ Gaussian filter to retain local structural information while mitigating the effect of noise for robust SSIM computation.

---

**Algorithm 1** Algorithm for Warm-Starting with Noisier2Noise (Moran et al., 2020)

---

**Require:** Noisy observations $\{y_i\}_{i=1}^N$, number of iterations $T$
**Ensure:** Partially trained denoiser $\hat{x}_\theta$
 1: Initialize network parameters $\theta$ randomly
 2: **for** $t = 1$ to $T$ **do**
 3:     Sample noisy batch $y$
 4:     Sample noisier batch $y'$
 5:     x $\leftarrow \hat{x}_\theta(y')$
 6:     Compute gradient on $\|x - y\|_2$
 7:     Update $\theta$
 8: **end for**
       **return** $\hat{x}_\theta$

---

**Algorithm 2** Gen-MURE Algorithm

---

**Require:** Noisy observations $\{y_i\}_{i=1}^N$
**Ensure:** Trained denoiser $\hat{x}_\theta$
 1: Initialize network parameters $\theta$ randomly
 2: **while** convergence **do**
 3:     Sample noisy batch $y$
 4:     $\hat{x}(y) \leftarrow \hat{x}_\theta(y)$
 5:     $K \leftarrow \frac{1}{k^2}\mathbf{1}_{k \times k}$
 6:     $\mu, \sigma^2 \leftarrow y * K, (y^2) * K - (y * K)^2$
 7:     $c \leftarrow \mathcal{N}(\mathbf{0}, \mathbf{I}_{m \times m})$
 8:     $\mathcal{L}_{\text{SSIM}} = \text{SSIM}(\hat{x}(y), y * c)$
 9:     $\mathcal{L} = \mathbb{E}\left[\|\hat{x}(y) - y\|^2 - \|y\|^2 \frac{\sigma^2}{\sigma^2 + \mu^2} + 2(y - \mu)^T \hat{x}(y)\right]$
10:     Compute gradient on $\mathcal{L} + \lambda \mathcal{L}_{\text{SSIM}}$
11:     Update $\theta$
12: **end while**
       **return** $\hat{x}_\theta$

---

To get noisy images $y$ corresponding to specific noise distributions during training and evaluation, training images were corrupted with speckle sampled from gamma distribution with varying number of looks and Rayleigh distribution having different scale parameters.

# 4 Experiments

## 4.1 Data

(i) We evaluated the proposed Gen-MURE method for real ultrasound datasets (Kaggle, 2016; Momot, 2022). (Kaggle, 2016) comprises images of the lymph nodes, brachial plexus and fetal heads, and each image is 256 $\times$ 320 pixels in dimension. (Momot, 2022) comprises ultrasound images of the common carotid artery of 11 volunteers. The image sizes range between 230 $\times$ 390 and 450 $\times$ 600 pixels.
(ii) To evaluate the generalizability of Gen-MURE, we used a clean 256 $\times$ 256 simulated image as shown in Figure 4. (iii) To evaluate generalizability, we assess the denoising performance of the pretrained Gen-MURE model on a non-multi-look Sentinel 1 image (Sinergise), 921 $\times$ 1421 pixels in dimensions. This image exhibits high speckle variance (Figure 5) and is used to evaluate Gen-MURE under domain shifts and different acquisition conditions.

## 4.2 Implementation Details

All models were trained on $128 \times 128$ noisy image patches. The framework was warm-started with Noisier2Noise (Moran et al., 2020) training methodology for 5 epochs. The stochastic gradient descent (SGD) algorithm was used to optimize the model with a learning rate of 0.005. Next, the Gen-MURE training objective was used to train the model for 100 epochs with a batch size of 256 images. The SGD optimizer was used with an initial learning rate of 0.005 which is reduced to 0.0005 after 50 epochs. All models were trained in three random training and validation partitions, where 70% of the data set was used for training and the remaining 30% was used to validate the trained model. The training images were augmented with random rotations and corruption with additive Gaussian noise having $\mu = \{0.25, 0.5\}$ and $\sigma^2 = \{0, 0.001\}$. The final augmented training sets comprise approximately 6000 images. The Gen-MURE method was trained with gamma noise with shape parameter $\alpha$ (Eq. 19) ranging between [2, 30] and Rayleigh noise having $\sigma$ (Eq. 20) between [0.1, 1], to incorporate a wide range of gamma and Rayleigh noise intensities for a thorough evaluation for the proposed method. For each noise distribution, the parameters $\alpha$ and $\sigma$ were linearly interpolated to generate a total of 200 noise levels for training and validation frameworks. A randomly sampled noise with a prespecified strength was first used to corrupt the input image. Next, the corrupted image is used to train Gen-MURE to estimate the denoised image without any access to the ground truth. For all experiments with Gen-MURE, $5 \times 5$ filters were used for SSIM and the SSIM hyperparameter was fixed as $\lambda = 1$.

## 4.3 Evaluation Metrics and Baseline Methods

We evaluated the performance of Gen-MURE with the structural similarity index measure (SSIM) and the peak signal-to-noise ratio (PSNR). Denoised images predicted by the model are compared to ground truth images. Higher values are desired for both PSNR and SSIM metrics. For ground truth and predicted images $I$ and $\bar{I}$, PSNR and SSIM scores are given as follows.

$$\text{PSNR} = 10 \log_{10} \Big( \frac{I_{MAX}^2}{|I - \bar{I}|^2} \Big); \quad \text{SSIM} = \frac{(2\mu_I \mu_{\bar{I}} + a)(2\sigma_{I\bar{I}} + b)}{(\mu_I^2 + \mu_{\bar{I}}^2 + a)(\sigma_I^2 + \sigma_{\bar{I}}^2 + b)} \tag{23}$$

$I_{MAX}$ is the maximum image intensity, $\mu_I$ and $\mu_{\bar{I}}$ denote the mean pixel intensities, $\sigma_I^2$ and $\sigma_{\bar{I}}^2$ denote the variances of pixel intensities, $\sigma_{I\bar{I}}$ represents the covariance between $I$ and $\bar{I}$ and $a$ and $b$ are constants.

We compared the performance of the proposed Gen-MURE framework with classical, self-supervised and diffusion model-based denoising algorithms, incorporating a broad spectrum of noise models and algorithmic strategies. This enables a thorough comparison comprising both classical and deep learning frameworks under different noise distributions. (i) **Speckle Reducing Anisotropic Diffusion (SRAD)** (Yu & Acton, 2002): The partial differential equation (PDE) based denoising algorithm was specifically aimed at reducing speckle. The algorithm iteratively smooths homogeneous regions and preserves fine structural details and edges, by modulating the diffusion coefficient based on the local pixel intensities. SRAD guides the diffusion process with the coefficient of variation and implements anisotropic diffusion, thus effectively reducing the speckle of corrupted images. (ii) **Block Matching and 3D Filtering (BM3D)** (Dabov et al., 2006): BM3D is a state-of-the-art denoising algorithm used for enhancing images from a wide class of applications. By grouping similar patches from noisy images, BM3D first obtains 3D stacks and subsequently performs collaborative filtering in a transfer domain. This effectively suppresses noise while preserving fine structural details. Although BM3D is not specifically designed for multiplicative speckle, it is highly robust and is used to enhance images from different applications. (iii) **Non-Local Means (NLMeans)** (Buades et al., 2005): NLMeans is a classical denoising algorithm and works by averaging pixels having similar neighborhoods. This is highly effective in reducing noise and preserving image textures. The algorithm works by exploiting non-local similarity of corrupted pixels and does not assume any explicit noise model. (iv) **Noisier2Noise (Nr2N)** (Moran et al., 2020): Nr2N is an unsupervised deep learning-based image denoising framework and can be used in scenarios where only noisy observations are available. Pairs of independent, noisy images are used for training the model and is used where clean ground truth images are unavailable. This is

highly relevant in denoising real-world noisy images, where only noisy observations can be obtained. (v) **Speckle2Void (S2V)** (Molini et al., 2022): S2V is a self-supervised deep learning framework specifically proposed to reduce speckle. This method uses the Noise2Void (Krull et al., 2019) framework and models speckle via the gamma distribution. It takes four rotated versions of the noisy image and uses a blind spot network to estimate the parameters of the inverse gamma distribution for each pixel. These estimated parameters are used to obtain the final denoised image. (vi) **Deep Image Prior (DIP)** (Ulyanov et al., 2018): DIP is a self-supervised framework for image restoration applications and takes advantage of the implicit bias that exists in convolutional neural networks. DIP initializes a convolutional network randomly, which is used as an implicit image model. With a random tensor as input, the output is generated by the convolutional network, the parameters of which are optimized for the input image by minimizing the loss between the generated and input images. (vii) **Noise2Self (N2S)** (Batson & Royer, 2019): N2S is a self-supervised image denoising method which assumes that noise is statistically independent along different measurement directions, unlike the clean signal underlying. The image features are partitioned into subsets with independent noise components. The learned denoising method can be used for enhancing images using single noisy observations and achieves denoising performance comparable to supervised algorithms. (viii) **Blind Diffusion Models (BlindDM)** (Li et al., 2026): BlindDM is a diffusion model (Ho et al., 2020) and is proposed for a supermeld-resolution of blind images. By integrating MAP-based optimization methods in diffusion models, the framework learns a joint distribution between low resolution and high resolution images along with degradation kernels and subsequently implements blind super-rsolution by coupling the unfolding of the MAP optimization with the reverse process of the diffusion model. Since the formulations rely on learning a robust image prior, these models are also applicable for blind image denoising. This method is used as a general-purpose diffusion-based image restoration framework.

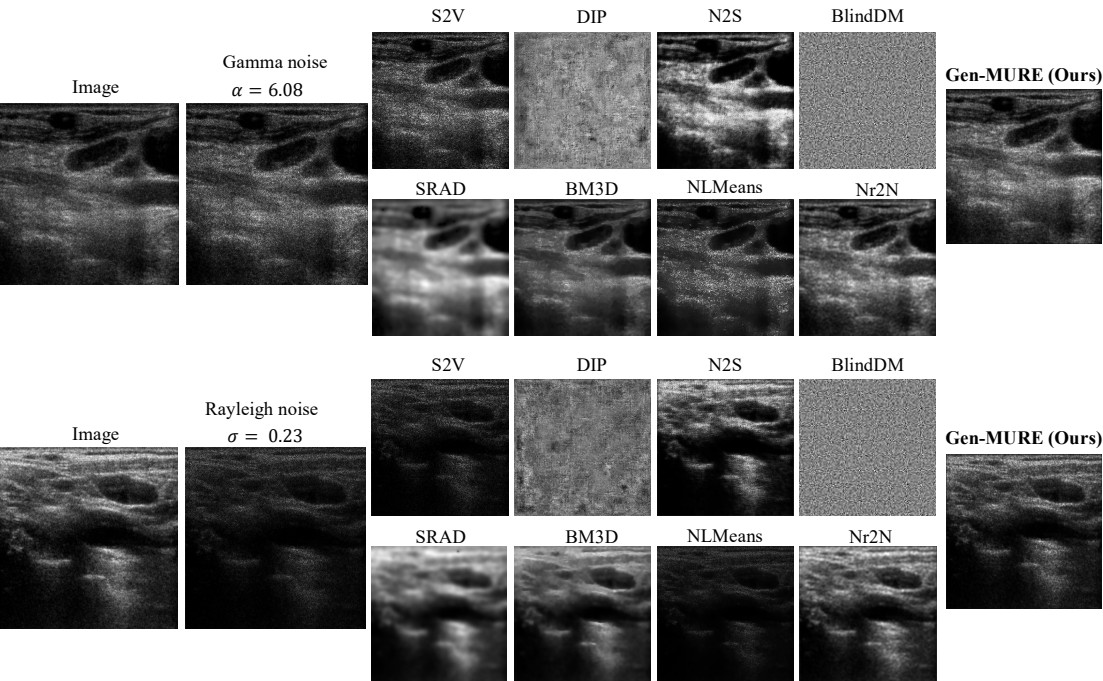

Figure 1: Columns 1 and 2 show the original and noisy images for gamma and Rayleigh noise corruption. Columns 3-6 show the images denoised by the baseline methods: (i) rows 1 and 2 show the results for gamma noise and (i) rows 3 and 4 show the results for Rayleigh noise. The last column shows the denoised images obtained with Gen-MURE for gamma and Rayleigh speckle.

### 4.4 Results

All methods were evaluated with different strengths of speckle having gamma and Rayleigh distributions. The evaluation scores for the ultrasound datasets are given in Table 1. For all experiments, we have reported the PSNR and SSIM scores for three different noise levels to correspond to different levels of speckle degradation for a comprehensive evaluation of the proposed method: (i) $\alpha = \{30, 15.93, 6.08\}$ for the gamma distribution and (ii) $\sigma = 1.00, 0.55, 0.23$ for the Rayleigh distribution. Figure 1 shows two images along with noisy versions, one corrupted with gamma noise ($\alpha = 6.08$) and the other with Rayleigh noise ($\sigma = 0.23$). We show the qualitative performance of all denoising methods in columns 3 to 7. We can see that, compared to all other methods, the denoised images obtained with Gen-MURE retain the fine structural details present in the original images. It is also validated numerically in Table 1 that Gen-MURE consistently performs better than all models for gamma noise and most models for Rayleigh distribution. The PSNR and SSIM scores for the images denoised by the proposed Gen-MURE remain consistently higher than most state-of-the-art image denoising methods. Even though gamma distribution is widely used to represent speckle in multilook SAR, which is associated with integer values, it should be noted that the gamma distribution is well-defined for positive real-valued shape parameters. Thus, we interpret $\alpha$ as the general gamma distribution shape parameter that determines the speckle characteristics, rather than strictly as a physical integer-valued number of looks.

Table 1: Comparison of different methods under gamma and Rayleigh noise distributions. Metrics reported are PSNR and SSIM for each noise level.

| Noise | Level | Metric | SRAD | BM3D | NLMeans | Nr2N | S2V | DIP | N2S | BlindDM | Gen-MURE (Ours) |
|---|---|---|---|---|---|---|---|---|---|---|---|
| gamma | $\alpha = 30.00$ | PSNR | 21.27 | 31.99 | 29.47 | 29.36 | 29.54 | 4.74 | 14.35 | 12.05 | **34.85** |
| | | SSIM | 0.700 | 0.853 | 0.849 | 0.835 | 0.882 | 0.195 | 0.443 | $0.152 \times 10^{-3}$ | **0.949** |
| | $\alpha = 15.93$ | PSNR | 21.17 | 31.68 | 26.92 | 28.89 | 27.43 | 4.77 | 14.36 | 11.88 | **33.53** |
| | | SSIM | 0.699 | 0.850 | 0.790 | 0.830 | 0.820 | 0.194 | 0.422 | $0.134 \times 10^{-3}$ | **0.933** |
| | $\alpha = 6.08$ | PSNR | 20.50 | 29.91 | 23.12 | 27.63 | 25.00 | 4.77 | 14.42 | 11.92 | **30.93** |
| | | SSIM | 0.696 | 0.826 | 0.690 | 0.815 | 0.694 | 0.194 | 0.379 | $0.111 \times 10^{-3}$ | **0.894** |
| Rayleigh | $\sigma = 1.00$ | PSNR | 21.38 | 24.36 | 19.18 | 22.25 | 20.02 | 3.43 | 17.95 | 12.08 | **25.34** |
| | | SSIM | 0.690 | 0.758 | 0.598 | 0.749 | 0.303 | 0.223 | 0.434 | $7.062 \times 10^{-5}$ | **0.823** |
| | $\sigma = 0.55$ | PSNR | 12.03 | 20.63 | 21.12 | 19.95 | **25.93** | 3.50 | 18.34 | 12.18 | 20.51 |
| | | SSIM | 0.649 | 0.750 | 0.631 | 0.747 | 0.430 | 0.223 | 0.431 | $8.669 \times 10^{-5}$ | **0.819** |
| | $\sigma = 0.23$ | PSNR | 0.50 | 4.14 | 13.59 | 5.86 | **22.94** | 3.73 | 18.32 | 12.08 | 7.24 |
| | | SSIM | 0.467 | 0.498 | 0.473 | 0.481 | 0.460 | 0.224 | 0.433 | $9.715 \times 10^{-5}$ | **0.525** |

Figure 2 shows the mean and variance of PSNR and SSIM scores for Gen-MURE, Noisier2Noise and Speckle2Void algorithms. These algorithms are compared since all these algorithms are data-driven neural network frameworks and have comparable performance across gamma and Rayleigh noise levels. All experiments were conducted thrice and the average scores are reported in Table 1.

Figure 3 shows a qualitative comparison of the denoising performance for gamma and Rayleigh multiplicative noise. The zoomed-in images highlight the efficiency of the proposed Gen-MURE framework towards recovering fine structural details even from severely degraded images. The edges, patterns and subtle intensity variations are well-preserved in the denoised images, even after training without ground truth images. This indicates that Gen-MURE can effectively reduce speckle while preserving high-frequency details in the images. This is in contrast to the state-of-the-art baseline methods which often oversmooth the fine structures or have artifacts in the denoised images.

In Figure 4, we show the simulated clean image and the noisy and denoised versions corresponding to different levels of gamma and Rayleigh speckle. The model trained on ultrasound dataset is used to denoise the noisy images. It is observed that even under severe corruption, the model can reduce speckle and preserve the edges for both noise distributions. Table 2 shows the PSNR and SSIM scores for different strengths of gamma and Rayleigh distributions. Figure 5 shows a noisy, real synthetic aperture radar image and the denoised versions obtained with Gen-MURE pretrained on gamma and Rayleigh distributions. It can be observed that, in both denoised images, the speckle is suppressed while preserving critical structural details.

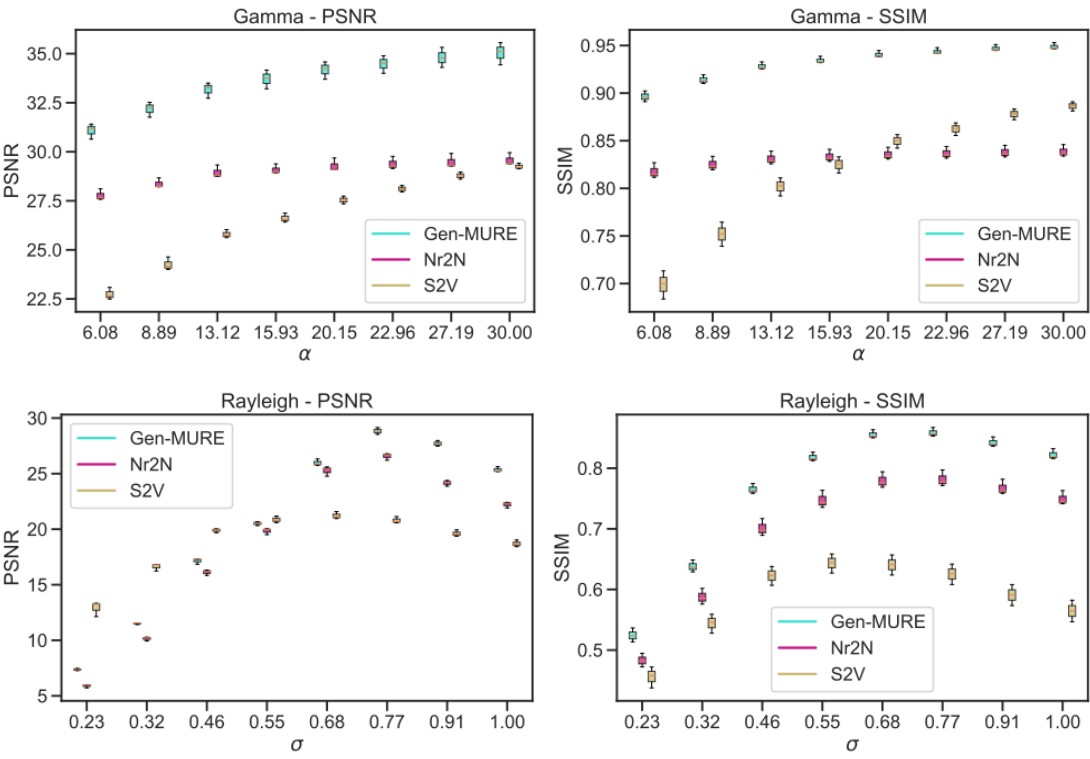

Figure 2: Mean and variance of PSNR and SSIM scores for Gen-MURE, Noisier2Noise and Speckle2Void.

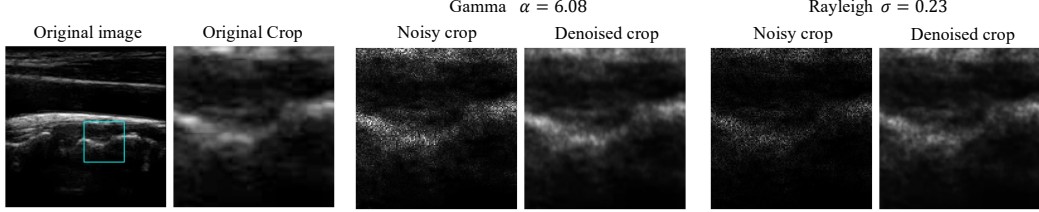

Figure 3: Corresponding to an ultrasound image, we zoom into a region. Next, we show the noisy and denoised versions for the cropped region for gamma and Rayleigh noise. The critical structures are retained in both the denoised images.

Table 2: Quantitative evaluation of simulated clean image for gamma and Rayleigh speckle

| gamma | | | | | | | | Rayleigh | | | | | | | |
|---|---|---|---|---|---|---|---|---|---|---|---|---|---|---|---|
| $\alpha = 8.89$ | | $\alpha = 15.93$ | | $\alpha = 22.96$ | | $\alpha = 30$ | | $\sigma = 0.32$ | | $\sigma = 0.55$ | | $\sigma = 0.77$ | | $\sigma = 1.0$ | |
| PSNR | SSIM | PSNR | SSIM | PSNR | SSIM | PSNR | SSIM | PSNR | SSIM | PSNR | SSIM | PSNR | SSIM | PSNR | SSIM |
| 26.23 | 0.677 | 28.08 | 0.732 | 29.08 | 0.768 | 29.80 | 0.791 | 7.43 | 0.548 | 15.11 | 0.598 | 21.45 | 0.567 | 21.66 | 0.525 |

## 4.5 Ablation Studies

We investigate the contribution of SSIM loss function and the warm start mechanism on the final evaluation scores. In Table 3 we compare the PSNR and SSIM scores for the proposed Gen-MURE method, as well as for ablated variants that remove the SSIM loss, the warm-start initialization, and both components simultaneously. We report the evaluation scores for three different noise levels for both distributions.

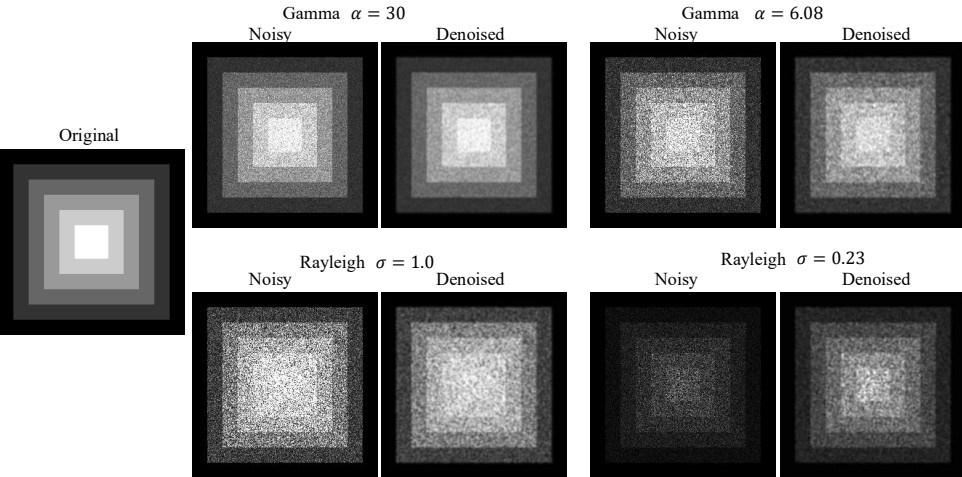

Figure 4: Column 1 shows the simulated image. Columns 2-5 show the noisy and denoised images for different strengths of gamma and Rayleigh speckle.

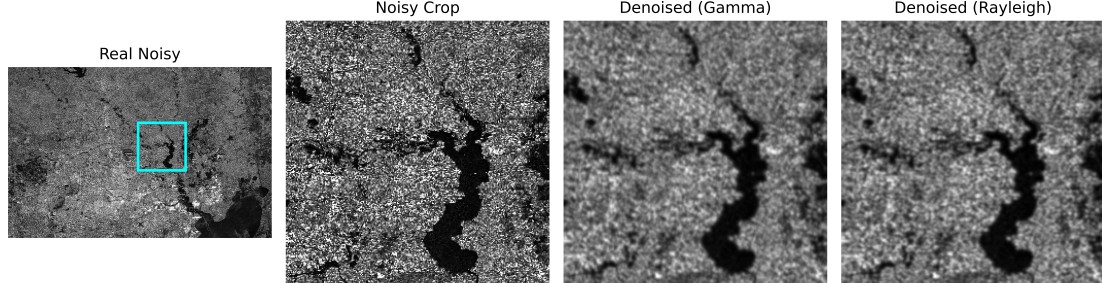

Figure 5: Column 1: Real SAR image corrupted with speckle. In columns 2-4, we show a noisy region along with the denoised images obtained with pretrained Gen-MURE with gamma and Rayleigh speckle respectively.

The PSNR and SSIM scores of Table 3 demonstrate that eliminating both SSIM loss and warm-start initialization leads to a drastic degradation in performance for all noise levels. When Gen-MURE is warm-started but trained without the SSIM loss, the denoising performance consistently deteriorates for all noise levels for both gamma and Rayleigh distributions. In contrast, incorporating the SSIM loss significantly improves the performance of Gen-MURE even in absence of warm-start initialization, demonstrating its critical role in effectively guiding the denoising performance in the absence of ground truth images. These results highlight the complementary strengths of the SSIM loss and the warm-start strategy and demonstrate their contributions to robust performance of the proposed method.

It should also be noted that even though the Noisier2Noise algorithm was proposed for reducing additive noise, it serves as an effective warm-start start method for multiplicative noise. This can be attributed to the fact that multiplicative noise can be expressed as additive noise in the logarithmic domain. The network may implicitly learn this mapping, thereby making Noisier2Noise an effective initialization strategy for signal-dependent, multiplicative speckle.

While the PSNR and SSIM scores for Gen-MURE w/o WarmStart and Gen-MURE are similar for all noise levels, Gen-MURE consistently achieves equal or higher performance scores. In particular, systematic improvements can be observed at several noise levels. These results indicate that the warm-start initialization

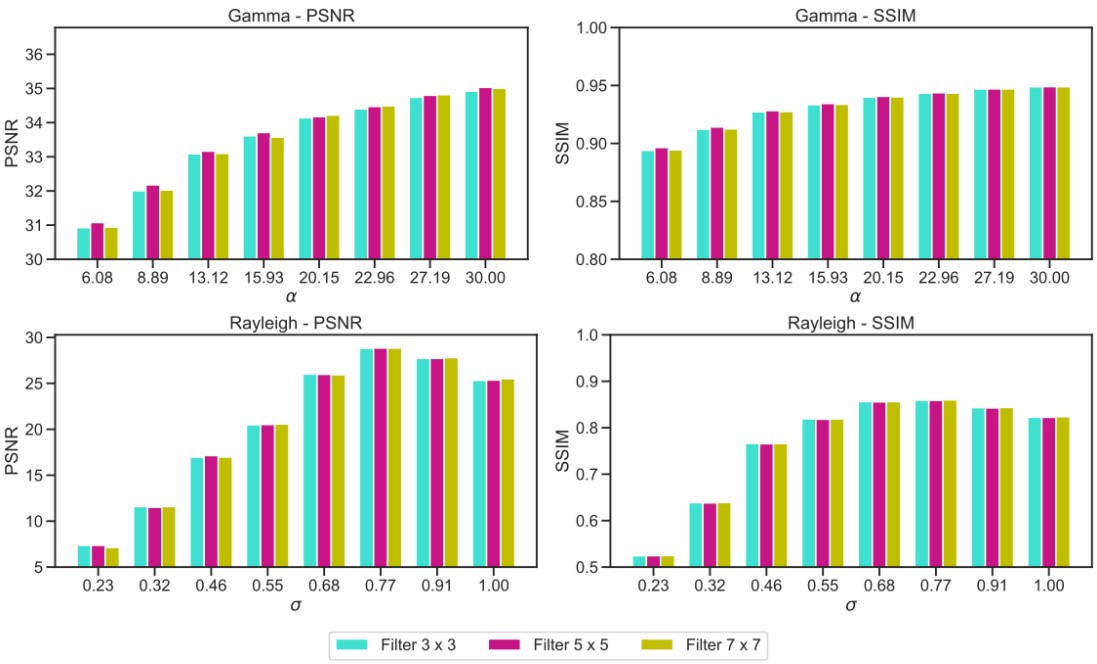

Figure 6: PSNR and SSIM scores for gamma and Rayleigh speckle for different filter sizes for SSIM implementation.

Table 3: Ablation studies for SSIM and WarmStart

| Method | gamma | | | | | | Rayleigh | | | | | |
|---|---|---|---|---|---|---|---|---|---|---|---|---|
| | $\alpha = 30.00$ | | $\alpha = 15.93$ | | $\alpha = 6.08$ | | $\sigma = 1.00$ | | $\sigma = 0.55$ | | $\sigma = 0.23$ | |
| | PSNR | SSIM | PSNR | SSIM | PSNR | SSIM | PSNR | SSIM | PSNR | SSIM | PSNR | SSIM |
| Gen-MURE w/o SSIM | 32.01 | 0.867 | 31.63 | 0.863 | 30.39 | 0.853 | 24.07 | 0.841 | 18.95 | 0.778 | 5.21 | 0.454 |
| Gen-MURE w/o WarmStart | 34.67 | 0.948 | 33.40 | 0.933 | 30.85 | 0.894 | 25.23 | 0.821 | 20.54 | 0.817 | 7.35 | 0.523 |
| Gen-MURE w/o SSIM w/o WarmStart | 19.78 | 0.552 | 19.77 | 0.551 | 19.38 | 0.545 | 10.21 | 0.446 | 6.92 | 0.421 | 2.17 | 0.383 |
| **Gen-MURE (proposed)** | **34.85** | **0.949** | **33.53** | **0.933** | **30.93** | **0.894** | **25.34** | **0.823** | 20.51 | **0.819** | 7.24 | **0.525** |

offers a stable and reliable performance gain, accelerating convergence across noise distributions and severity levels.

In Figure 6, we study the variations in final PSNR and SSIM scores when the filter sizes are varied. All evaluations for Table 1 were reported for $5 \times 5$ filters. We additionally evaluate the denoising performance on two more filter sizes, $3 \times 3$ and $7 \times 7$. It is observed that the $5 \times 5$ filter gives highest PSNR and SSIM scores for gamma noise. For Rayleigh noise, both PSNR and SSIM scores are almost identical. Figure 7 demonstrates the effect on PSNR and SSIM scores when the SSIM hyperparameter $\lambda$ is varied. We evaluate the PSNR and SSIM scores for $\lambda = \{0.5, 1, 1.5\}$. It can be observed that in all cases, the best scores are obtained for $\lambda = 1$.

# 5 Discussions

## 5.1 Computational Complexity

Table 4: Comparison of Speckle2Void and Gen-MURE models in terms model complexity.

| Speckle2Void | | | Gen-MURE | | |
|---|---|---|---|---|---|
| Total params | GFLOPS | Peak Memory Usage (MB) | Total params | GFLOPS | Peak Memory Usage (MB) |
| $1.5 \times 10^5$ | 2.46 | 65.14 | $2.2 \times 10^5$ | 3.64 | 33.23 |

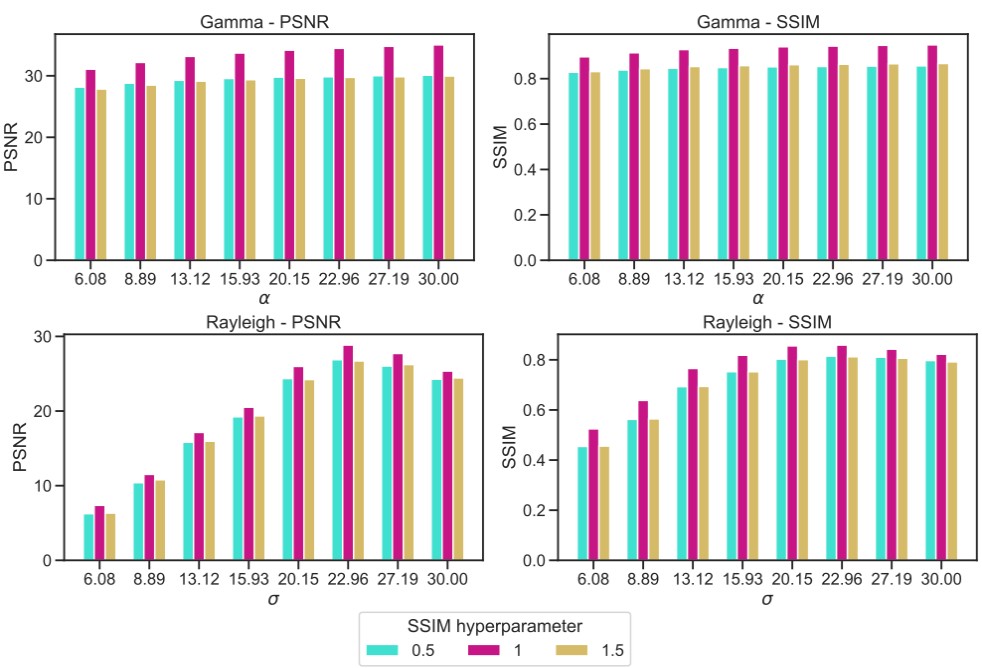

Figure 7: PSNR and SSIM scores for gamma and Rayleigh speckle for different values of SSIM hyperparameter values.

In Table 4, we compare the computational complexities of Speckle2Void (Molini et al., 2022) and the proposed Gen-MURE framework. Among all state-of-the-art image denoising baseline models, Speckle2Void is the only method specifically designed for reducing speckle without requiring clean ground truth images. However, Speckle2Void is limited by its explicit assumption that speckle follows a gamma distribution, whereas Gen-MURE is a general framework applicable to a broad class of multiplicative noise models. Additionally, Speckle2Void takes four rotations for any input image and estimates the parameters of the inverse gamma distribution for each pixel. This makes the overall method computationally expensive and significantly increases memory consumption. This can be validated from Table 4. Although Gen-MURE contains slightly more learnable parameters, it achieves significantly lower peak memory usage and avoids repeated rotational evaluations, resulting in a more computationally efficient and scalable solution. This makes Gen-MURE particularly well-suited for large-scale and resource-constrained deployment scenarios.

## 5.2 Generalizability

For deploying denoising algorithms for real-world imaging systems, it is extremely critical that the model generalizes across acquisition settings and different noise statistics. In many practical scenarios, the underlying corruption model is unknown and often varies across sensors and acquisition conditions. Denoising methods that assume particular noise distributions often have limited generalization capabilities.

To evaluate the generalization capabilities of Gen-MURE, we used models trained on ultrasound images with gamma and Rayleigh distributions to denoise unseen test images with significantly different image structures and obtained from different sensing modalities. Specifically, we consider two challenging out-of-distribution scenarios (i) a simulated image comprising sharp edges and corners (Figure 4) and (ii) a noisy real SAR image (Figure 5). The denoised images show that the underlying structures are well-preserved, along with the high-frequency details, thus highlighting the efficiency of Gen-MURE in operating under distribution and modality shifts. Gen-MURE does not assume that the speckle follows any particular distribution, which enables it to suppress multiplicative noise across diverse imaging domains, demonstrating its robustness and generalization capabilities.

### 5.3 Limitations

The proposed method Gen-MURE has three potential limitations. Tables 1 and 3 demonstrate that the warm start strategy with Noisier2Noise (Moran et al., 2020) consistently improves the denoising performance for both gamma and Rayleigh distributions in terms of SSIM scores. It also improves the PSNR scores across all noise levels for the gamma distribution and for most noise levels for the Rayleigh distribution. Even though Noisier2Noise was primarily proposed for additive noise, we observe that it can also serve as an efficient initialization strategy for reducing multiplicative noise. However, there is a potential limitation to this approach. The features extracted during initial training stages are propagated through the later stages of optimization. Consequently, any oversmoothing or residual artifacts introduced during the warm start initialization may persist and can lead to artifacts in the final denoised images. This can influence the efficiency of the overall denoising algorithm, particularly in images with severe speckle corruption.

Next, Gen-MURE assumes the speckle to have a unit mean. This assumption might be violated for some coherent imaging applications. It can be validated from Table 1 that Gen-MURE can efficiently reduce speckle from images even when the mean deviates from unity for Rayleigh noise. For $\sigma = \{1, 0.55, 0.23\}$, the mean is never exactly unity. However, it should be noted that in all these cases, Gen-MURE can effectively reduce speckle from the corrupted images and can consistently recover the underlying clean image. These observations demonstrate the robustness of Gen-MURE and its ability to adapt to real world denoising applications even when the unit mean assumption is not strictly satisfied.

Finally, Gen-MURE assumes that the unknown noise $\epsilon$ is sampled from some distribution and does not consider the scenario where $\epsilon$ varies spatially. In such a scenario, the Gen-MURE formulations can lead to suboptimal denoising performance. For applications involving spatially varying unknown noise, an additional noise estimation step can be first used to estimate the noise statistics $\mathbf{E}[\epsilon^2]$ and $\mathbf{E}[\epsilon]$. This can be followed by the training objective for Gen-MURE (Eq. 14). Alternating between noise estimation and denoising may improve the ability of the overall framework to reduce speckle from images corrupted with spatially varying noise.

## 6 Conclusion

This work presents a model-agnostic self-supervised speckle-reducing framework **Gen-MURE**. In contrast to prior approaches, Gen-MURE does not require the underlying noise distribution to be known and generalizes across a wide range of imaging modalities and noise distributions. Extensive experiments on ultrasound, simulated and real SAR images demonstrate that Gen-MURE efficiently preserves critical structures while suppressing noise following gamma and Rayleigh distributions. Speckle in single-look SAR images can be modeled with the exponential distribution. Since exponential distribution is a special case of gamma distribution, Gen-MURE can also be used to reduce speckle from single-look SAR images without requiring any modifications to the original formulation. Additionally, Gen-MURE has higher computational efficiency and lower memory usage, compared to other self-supervised speckle reducing baselines, thus making it highly robust and suitable for real-world deployment.

### Broader Impact Statement

Image denoising algorithms for coherent imaging modalities can significantly boost the performance of downstream tasks. However, in some cases the denoised images may exhibit oversmoothing and residual artifacts leading to inconsistent structural representations or suppression of clinically relevant features. Thus caution should be exercised when this algorithm is deployed for real-world image enhancement.

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
