# OpenReview forum: "Gen-MURE: Generalized Multiplicative Unbiased Risk Estimate"
_TMLR — Decision pending for TMLR_

### Review · Reviewer_9xdC · 2026-03-14

**Summary Of Contributions:**

The paper introduces Gen-MURE, a self-supervised image denoising framework designed for signal-dependent multiplicative noise (speckle) commonly found in coherent imaging systems like ultrasound and SAR. Unlike existing methods, Gen-MURE is model-agnostic; it does not require explicit knowledge of the noise distribution parameters or access to clean ground-truth images. The authors derive a generalized unbiased risk estimate and enhance the training stability using a modified SSIM loss and a Noisier2Noise warm-start strategy.

## Strengths:

- Theoretical Grounding: The derivation of the generalized objective from the standard MURE formulation is elegant and removes the strict reliance on known noise parameters.
+2


- Computational Efficiency: Compared to state-of-the-art self-supervised speckle reduction methods like Speckle2Void, Gen-MURE demonstrates significantly lower peak memory usage and computational overhead.


- Empirical Robustness: The method shows consistently strong performance across both Gamma and Rayleigh noise distributions on real ultrasound and SAR data.

## Weaknesses:

- Heuristic Surrogate: The modified SSIM loss relies on a simple m×m Gaussian filter as a local smoothing operator to approximate the clean image. This is somewhat heuristic and could introduce blurring artifacts.


- Generalization Scope: While claimed as "generalized," the empirical validation is strictly bounded to Gamma and Rayleigh distributions.

**Audience:**

Yes

**Audience Explanation:**

The paper tackles a highly practical problem in computational imaging. Researchers and practitioners working with remote sensing (SAR) and medical imaging (ultrasound) will find a computationally efficient, self-supervised speckle reduction method highly valuable, especially since acquiring clean ground-truth data in these domains is often impossible.

**Broader Impact Concerns:**

There are no significant negative ethical implications associated with this work. Improving image quality in medical and satellite imaging is broadly beneficial. However, as with any medical image enhancement algorithm, a standard disclaimer should be considered: the model risks hallucinating structures or over-smoothing subtle, clinically relevant anomalies. A brief acknowledgement of this in a Broader Impact Statement would suffice.

**Claims And Evidence:**

Yes

**Claims Explanation:**

The theoretical derivations for the generalized risk estimate are mathematically sound and clearly presented. The empirical evidence is convincing: the authors benchmark Gen-MURE against a solid mix of classical (BM3D, SRAD), self-supervised (Noisier2Noise, Speckle2Void), and diffusion-based (BlindDM) baselines. Furthermore, the ablation studies successfully isolate and validate the individual contributions of the SSIM loss and the warm-start mechanism.

**Requested Changes:**

## Critical to securing recommendation:

- Sensitivity of the Smoothing Operator: The SSIM loss heavily relies on the m×m Gaussian filter c(y). Please include an ablation or discussion on the sensitivity of the hyperparameter m. How does the kernel size impact the preservation of high-frequency structural details versus noise suppression?


- Assumption Justification: The derivation assumes a unit mean for the multiplicative noise (E[ϵ]=1). Please briefly discuss the limitations of this assumption. Are there coherent imaging scenarios where this does not hold, and how would Gen-MURE adapt?

## To strengthen the work:
- Spatially Varying Noise: Briefly discuss how Gen-MURE performs (or could be extended) if the noise distribution or intensity is not globally stationary but varies spatially across the image.

- Limitations Section: Add a dedicated paragraph detailing the failure modes of the Noisier2Noise warm-start strategy, as the ablations show the model heavily relies on it for optimal performance.

---

> ### Author Response · Authors · 2026-05-28
> **Responses to Reviewer 9xdC**
>
> We thank the reviewer for the comments.
> 1. Sensitivity of the Smoothing Operator: Fig 6 of the revised manuscript shows ablation study for the final PSNR and SSIM scores obtained when the $m \times m$ filter sizes are varied as $3 \times 3$, $5 \times 5$ and $7 \times 7$. Smaller kernel size retains more speckle whereas higher kernel size oversmooths the image. This can be observed from the PSNR and SSIM scores for both noise distributions.
> 2. Assumption Justification: Although the formulations assume a unit mean, Gen-MURE can perform desirably even when the mean deviates from 1. This can be validated by the performance of Gen-MURE for Rayleigh noise. The Rayleigh distribution has a mean of $1.253\sigma$ and in Table 1, we have considered $\sigma = {1, 0.55, 0.23}$. Thus, in none of these cases, the unit mean assumption holds. Even then it can be observed that Gen-MURE can denoise the noisy images desirably. Additionally, for most of these cases, Gen-MURE performs significantly better than all state-of-the-art baseline methods. We have added this discussion in Section 5.3 of the revised manuscript.
> 3. Spatially Varying Noise: Gen-MURE assumes that the noise is from some unknown distribution and is multiplicative in nature. However, this work assumes that the noise is iid, which simplifies the term $\| x \|^2 = \dfrac{\|y\|^2}{\mathbf{E[\epsilon^2]}}$. This simplification might not hold if the noise is spatially varying across the image. To address that scenario, an additional noise estimation step (such as a mixture of Gaussians) can be used to first estimate the noise statistics $\mathbf{E}[\epsilon^2]$ and $\mathbf{E}[\epsilon]$, and then can be followed by the proposed training objective (Eq. 14 of the revised manuscript). We have added this discussion in Section 5.3 of the revised manuscript.
> 4. Table 3 shows that there is an improvement in most cases when both SSIM and WarmStart are used, compared to when just SSIM (and no WarmStart) is used (row 2 and row 4). However, if the WarmStart strategy introduces oversmoothing or some residual artifacts, the undesired effects will propagate throughout the training process. This can degrade the overall performance of the proposed method. We have added this discussion in Section 5.3 of the revised manuscript.

---

### Review · Reviewer_bswi · 2026-05-13

**Summary Of Contributions:**

This paper introduces the Gen-MURE image denoising algorithm, which provides a novel approach to denoising images that contain multiplicative, signal-dependent noise. This characteristic of the noise is what makes the algorithm novel and important.

The proposed framework bridges two types of denoising approaches and works with a type of noise which is closer to real-world scenarios; and it illustrates the proposed framework with a complete set of experiments.

The two weaknesses of the work come from a lack of demonstrated experimental robustness (the experiments have not been run multiple times, which would illustrate the robustness of the method); and from the claim of not needing to know the distributions from which the noise of the images are sampled: this seems contradictory to the implementations made, where noise is sampled from a given distribution.

**Additional Comments:**

It would be valuable to release the code for the experiments, to allow readers to reimplement the studies proposed by the authors.

**Audience:**

Yes

**Audience Explanation:**

This paper proposes a novel denoising algorithm (Gen-MURE) which aims to overcome the weaknesses of the existing two families of denoising algorithms presented in the related works: the SURE family where the noise comes from a known distribution; and the self-supervised Noise2x frameworks which do not make this assumption but tend to perform less well. Both families of algorithms focus on additive, independent noise, which is often a strong assumption to make, and which the Gen-MURE algorithm addresses. I believe this addresses both the criteria of containing "new algorithms with sound empirical validation, optionally with justification of theoretical, psychological, or biological nature"; and of the "development of new analytical frameworks that advance theoretical studies of practical learning methods".

**Broader Impact Concerns:**

No Broader Impact Statement is provided, and there are no specific ethical implications which come to mind as needing to be cited.

**Claims And Evidence:**

Yes

**Claims Explanation:**

The authors claim to extend existing denoising frameworks in a principled way and to illustrate the effectiveness of this framework through an extensive set of experiments, which they do.

The framework is developed throughout Section 4, and experiments are presented in Section 5.

The authors claim that their approach brings the novelty of a generalised approach for settings with an unknown noise distribution. However, they pre-train a neural network to warm start their algorithm with the Noisier2Noise framework, which requires a known noise distribution; corrupt images to correspond to specific distributions (page 7); and evaluate the framework for specific levels of noise, given parameter distributions. Whilst this is not an issue in itself, maybe it would be worth nuancing the claims made at the start of the paper to reflect the specificities of the empirical choices made.

It would also be important to mention whether the experiments have been rerun multiple times, so as to be able to comment on the variability of the approach.

**Requested Changes:**

I have four sets of questions/suggestions: the text and the experiments each have a set of questions which are on form, and a set of question which go deeper on the contents. I believe that addressing these points would contribute to making the work more readable, as it would place it into a clearer context and make it more robust to criticisms.

Small typos, which would simply make the work easier to read.
- I suggest using the \mathbb{} command for $\mathbb{R}$ and $\mathbb{E}$, which tends to the convention.
- The sum at the bottom of page 3 is indexed by $i$, which I believe goes to $N$: make sure to specify what the sum goes to.
- Just before Equation 11 on page 5, I would use ``for all'' in English rather than the $\forall$ symbol, since it is outside of a block of mathematics.

Changes in form which would contextualise the approach more clearly and guide the reader clearly through its novelty.
- I would suggest reworking Sections 2 and 3 into a single section. This will make the mathematical formulation go hand-in-hand with the existing literature, rather than discussing the same approaches twice from two perspectives.
- The choice of metrics may be standard in the image denoising community, but would make the paper more readable and stand-alone to recall them at the start of the experimental section.
- I understand that SAR images each have a number of looks, which impacts the level of noise, and for the choice of distribution to model the nose. I believe it is possible to model single-look SAR images with noise sampled from the exponential distribution. Maybe the current paper is much more sophisticated and don't have the need to run experiments with the exponential distribution with the given dataset; but it would be a nice piece of background to mention this, and any methods that have used this distribution.
- The transition from Equation 12 to Equation 13 is not entirely clear to me, could you add a couple of lines to make this easier to understand?
- Subsection 4.2 nicely gives intuition for how your parameters $\alpha$ and $\sigma$ impact the Gamma and Rayleigh distribution. Can you give similar context for the choice of evaluated parameters in your Subsection 5.4? Additionally, can you explain why you have non-natural numbers for $\alpha$, if these are the to be thought as the number of looks in Subsection 4.2?

I am convinced by the quality of the experiments and that they validate the soundness of the approach. Answering this first set of questions would set more context for your experimental choices, and make the work more robust to criticism.
- Were your experiments run multiple times? Providing a mean and variance for your results would go further in highlighting the strength of your approach compared to your competitors.
- In Table 1, the Speckle2Void approach beats Gen-MURE for a Rayleigh distribution with $\sigma = 0.23$. However, the qualitative sample you provide in Figure 1 for this experiment shows that S2V has greatly changed in hue, almost as if colours has been inverted. It would be interesting to comment on this sample, as well as to highlight any shortcomings from the choice of metric which this showcases (since it is coming out as the best PSNR despite being qualitatively far away from the sample).
- Can you say more on how $y^\prime$ is sampled to train the warm-start algorithm, are you simply noising with a known distribution (which seems contrary to the claim made whereby no )?
- How did you select the network architecture for denoising? Are there other architectures which you could have chosen?

My second set of experimental questions could suggest a few possible further ablation studies, which could give the reader more intuition as to how each empirical choice enabled a better-performing algorithm.
- Equations 20 and 21 include a smoothing operator in the SSIM. It is worth briefly discussing what it is (how is the average taken, how is the number of cells selected...), which will lead the reader to understand if another smoothing operator could have been selected. This naturally paves the way to a possible set of ablation studies which could not only look at the effect that varying $m$ might have, but also look at the choice of loss and its impact: could we add a regularisation hyperparameter in front of the SSIM to see how its effect can be moderated? Could we use PSNR instead of SSIM?
- The warm-start Algorithm 1 comes from the Noisier2Noise paper. I would recommend adding context on this choice approach for someone who might not be as familiar: have other papers attempted warm-start trainings? Would there be other possible choices of warm-start algorithms? This would place the warm-start approach in the landscape of existing works and give intuition as to its innovation with respect to existing works.

---

> ### Author Response · Authors · 2026-05-28
> **Responses to Reviewer bswi - Part 1**
>
> We thank the reviewer for the comments. We have fixed all the typing errors and combined Sections 2 and 3, as suggested by the reviewer. We have added the formulations of PSNR and SSIM metrics at the beginning of Section 4.3.
>
> Exponential distribution: Since the exponential distribution is a special case of gamma distribution, our method can be also directly used to reduce speckle from single-look SAR images. We have added this context to Section 6 of the revised manuscript.
>
> We have added the steps between Eq 12 and Eq 13 of the original manuscript. We apologize for the inconvenience.
>
> Choice of parameters: The Gen-MURE method was trained with gamma noise with shape parameter $\alpha$  ranging between [2, 30] and Rayleigh noise having $\sigma$  between [0.1, 1], to incorporate a wide range of gamma and Rayleigh noise intensities for a thorough evaluation for the proposed method. We have reported the PSNR and SSIM scores for three noise levels selected from this range for both distributions. These noise levels correspond to low, medium, and high noise regimes, thus providing a comprehensive assessment of the denoising algorithm.
>
> We varied the shape parameter from 2 to 30 to evaluate the denoising performance on a wide range of shape parameters to represent the multilook scenario. To improve training diversity, the shape parameter was varied in over 200 steps, resulting in non-integer values of $\alpha$. This effectively provided a continuum of noise levels during training and implicitly augmented the data, thus further improving the robustness and generalization of the final model.
>
> While the multilook scenario is associated with integer values, the gamma distribution is well-defined for positive real-valued shape parameters. Thus, we interpret $\alpha$ as the general gamma distribution shape parameter that determines the speckle characteristics, rather than strictly as a physical integer-valued number of looks.
>
> We have added these in Section 4.2 and 4.4 of the revised manuscript.
>
> First set of questions:
>
> 1. Were your experiments run multiple times?:
> Each experiment was carried out three times and the mean scores are reported for both PSNR and SSIM. We have included a plot for visualizing mean and variance for Gen-MURE, Noisier2Noise and Speckle2Void since these methods have best scores for both PSNR and SSIM. Figure 2 in the revised manuscript shows the mean and variance of the aforementioned methods.
>
> 2. Speckle2Void evaluation:
> We sincerely apologize for this. We had mistakenly added the denoised image from a different experiment. We have updated the denoised results for Speckle2Void. The denoised images obtained for Rayleigh noise with Speckle2Void for $\sigma = 0.23$ retained the speckles, which led to higher PSNR and lower SSIM scores.
>
> 3. $y'$ for warm start:
> To train the warm start algorithm, we are noising the same image with different levels of noise (sampled from the same distribution). $y$ and $y'$ are noisy images of the same noise distribution, where $y'$ is noisier than $y$. Since this is not required during inference (denoising in real-world applications), this does not contradict the claims. Once the training is complete, the knowledge of the exact noise distribution is not required, as demonstrated by Fig. 5 of the revised manuscript (exact noise distribution is unknown; only noisy images are given). Both gamma and Rayleigh models can efficiently reduce speckle from the noisy image, while retaining the high-frequency details.
>
> 4. Network architecture:
> We selected the DnCNN architecture for its efficient architecture comprising CNN blocks and residual connections, which has been used extensively for image denoising applications. The residual learning framework leads to faster convergence while maintaining computational efficiency. In addition, we observed that DnCNN performed well even with limited training data.
> There are a few other architectures that could be explored, such as UNet-based backbones and transformer architectures. We intentionally avoided using transformer-based baselines that require large training datasets. Since the primary objective of the paper is to demonstrate the effectiveness of the self-supervised speckle reducing formulation, DnCNN was used, which provided a balance between performance, model complexity and training efficiency.

---

> ### Author Response · Authors · 2026-05-28
> **Responses to Reviewer bswi - Part 2**
>
> Second set of experimental questions:
> 1. SSIM:
> We apologize for the missing information. We used the $5 \times 5$ filter for all experiments.
> Fig 6 in the revised manuscript shows the effect of varying the filter sizes as $3 \times 3$, $5 \times 5$ and $7 \times 7$. It can be observed that the $5 \times 5$ filter gives the best evaluation scores, which can be attributed to its balance between speckle reduction and detail preservation.
>
> As recommended, we have added a hyperparameter $\lambda$ in front of the SSIM expression and studied the variations in final PSNR and SSIM scores as $\lambda$ is varied. We considered three values, $\lambda = \{0.5, 1, 1.5\}$ and the results are reported in Fig 7 of the revised manuscript.
>
> Calculating the PSNR explicitly depends on the mean squared error between the denoised image and the ground truth image. When the ground truth images are not available, the exact values cannot be calculated. Thus, we decided to use the SSIM scores.
>
> 2. Warm start strategy:
>
> Warm start strategies are used in several image analysis algorithms to improve the stability and accelerate convergence during early stages of training. In the context of self-supervised training, this becomes important since clean, noiseless images are not accessible to the training paradigm. The Noisier2Noise warm start strategy provides a simple and computationally efficient self-supervised algorithm that does not require access to two independent noisy images corresponding to the same original image. This is a simple algorithm where independent noisy realizations are obtained for any image by adding synthetic noise, making it a suitable choice for our framework. We have added this in Section 3.3.2.

---

### Review · Reviewer_e9H4 · 2026-05-16

**Summary Of Contributions:**

This paper proposes Gen-MURE, a self-supervised denoising framework for coherent imaging (e.g., ultrasound, SAR) corrupted by multiplicative noise. The method operates without clean ground-truth images or precise noise distribution parameters. It approximates risk estimation terms using the local mean and variance of the observed image, and trains the network using a Noisier2Noise warm-start strategy combined with a modified local SSIM loss.

**Additional Comments:**

The paper is well-structured with clear motivation. While the theoretical breakthrough is incremental, the overall pipeline is complete, the experimental validation is solid, and the method offers good practical value due to its low computational complexity.

**Audience:**

Yes

**Audience Explanation:**

Self-supervised denoising for medical and remote sensing imagery (where ground truth is typically unavailable) is a practical bottleneck. This work is relevant to researchers and practitioners in computer vision and applied machine learning.

**Broader Impact Concerns:**

No significant negative social impacts. However, given the application to medical imaging (ultrasound), care must be taken to ensure the denoising process does not inadvertently erase minute but critical pathological features due to local smoothing. I recommend adding a brief disclaimer regarding medical diagnostic risks.

**Claims And Evidence:**

Yes

**Claims Explanation:**

The experiments cover Gamma and Rayleigh multiplicative noise distributions. The paper demonstrates performance advantages over classical and deep learning baselines (e.g., BM3D, Speckle2Void) on both ultrasound and SAR datasets. Furthermore, the ablation studies adequately validate the necessity of the modified SSIM loss and the warm-start mechanism.

**Requested Changes:**

1. Theoretical limitations: Equation (16) strictly relies on the assumption that the multiplicative noise has a unit mean ($E[\epsilon]=1$) in coherent imaging. Please discuss the robustness of Equation (17) if real-world noise deviates from this assumption.

2. Warm-start justification: As shown in Table 3, the warm-start mechanism stabilizes performance. However, Noisier2Noise was originally designed for additive noise. Please briefly explain in the text why an additive-noise-based framework serves as an effective initialization for a multiplicative noise model.

3. Reframing novelty: The core theoretical derivation is relatively straightforward (substituting parameter terms in the original MURE with local statistics). I suggest toning down the claims of theoretical innovation in the introduction and instead emphasizing the practical robustness and engineering value of the complete pipeline.

---

> ### Author Response · Authors · 2026-05-28
> **Responses to Reviewer e9H4**
>
> We thank the reviewer for the comments.
>
> 1. Theoretical limitations:
> Although the formulations assume a unit mean, Gen-MURE can perform desirably even when the mean deviates from 1. This can be validated by the performance of Gen-MURE for Rayleigh noise. The Rayleigh distribution has a mean of $1.253\sigma$ and in Table 1, we have considered $\sigma = {1, 0.55, 0.23}$. Thus, in none of these cases, the unit mean assumption holds. Even then it can be observed that Gen-MURE can denoise the noisy images desirably. Additionally, for most of these cases, Gen-MURE performs significantly better than all state-of-the-art baseline methods. We have added this discussion in Section 5.3 of the revised manuscript.
>
> 2. Warm-start justification:
> Noisier2Noise, although originally proposed for additive noise, can also effectively reduce speckle from corrupted images (Table 1 - Nr2N). During the warm-start stage, Noisier2Noise learns coarse image features from speckle-corrupted images, which thereafter helps Gen-MURE achieve effective speckle reduction during the later stages of training.
>
> One possible explanation for the effectiveness of Noisier2Noise in multiplicative denoising is that multiplicative noise can be expressed as additive noise in the logarithmic domain. Consequently, the network may implicitly learn this transformation during the initial training stages, thereby making Noisier2Noise an effective warm-start initialization strategy even for multiplicative speckle noise.
>
> We have added a brief explanation in Section 4.5 of the revised manuscript.
>
> 3. Reframing novelty:
> We thank the reviewer for the suggestion. We have rephrased the novelty in Section 1 of the revised to emphasize on the practical applicability of the proposed method.

---

### Decision · Action_Editor_Q5ks · 2026-06-30

**Recommendation:** Accept with minor revision

**Additional Comments:**

Although reviewers found the methodological contribution and performance interesting, they had various concerns regarding the paper.
In particular on the assumptions (multiplicative noise with a unit mean)  suggesting to highlight more the practical robustness than the theoretical innovation. They also asked for additional runs to assess the statistical efficiency and required additional comments and justifications of the experiments (network architecture for denoising, ablation studies).
The authors mostly answered satisfactorily to these remarks by adding comments in various sections and by stregntening the numerical section. Overall, this provides an intersting practical contribution although I suggest some minor modifications in the manuscript.

-Some typos could be corrected after a careful proofreading (for instance between (10) et (11) I guess this should be (\epsilon_i)_{i=1}^N instead of \epsilon_{i=1}^N. So I suggest that following Reviewer  bswi remarks, you go through all the paper to edit all remaining typos.
-The fact that x is deterministic should be highlighted at the beginning of section 3 when the model is presented. Since the noise is state dependent, the fact that x is deterministic is crucial for the computation of expectations etc.
-I believe that the statement of the proposition is not precise enough. You propose an approximation of \mathbb{E}[\epsilon^2] but without any additional explanation we do not know if this approximation is sharp in the way the Proposition is written. I believe it would be more precise to provide in the Proposition the value of \mathbb{E}[\epsilon^2] without referring to an "approximation". This proposition  is simply an intermediate step to design the training objective and maybe the training objective could be more highlighted than the way the Proposition is written.
- I do not know if the experiments are computationally intensive but each experiment was carried out three times which seems low to assess variability of the results and I would suggest to run additional experiments to provide better guarantees.

**Audience:**

Yes

**Audience Explanation:**

The proposed model and algorithm introduce a new framework inspired by  two denoising approaches and propose numerical evaluations using  realistic noise distributions.  The paper addresses a practically important problem, i.e. image denoising when clean ground-truth images and exact noise distributions are not unavailable.
The highlight interesting performance with both Gamma and Rayleigh noise distributions on real ultrasound and SAR data.
For these reasons, this paper is of interest to a large audience in applied machine learning.

**Claims And Evidence:**

Yes

**Claims Explanation:**

The authors introduce Generalized Multiplicative Unbiased Risk Estimate, Gen-MURE, a framework for image denoising framework when observations are corrupted with multiplicative noise. They highlight that they do not use any parametric assumptions about the underlying noise in the derivation of the training process.

They evaluate the performance of Gen-MURE for specific  gamma and Rayleigh distributions and run experiments on ultrasound images to validate the denoising robustness of their algorithm compared to state-of-the-art image denoising baselines.

The experiments and evidence proposed by the authors allow to support their claims and highlight the novelty and performance of Gen-MURE even if reviewers proposed to slightly modify these claims (use of warm start, evaluation of the framework for specific levels of noise, etc.).